# The Link Between Paraquat and Demyelination: A Review of Current Evidence

**DOI:** 10.3390/antiox13111354

**Published:** 2024-11-05

**Authors:** Renata Silva, Ana Filipa Sobral, Ricardo Jorge Dinis-Oliveira, Daniel José Barbosa

**Affiliations:** 1Associate Laboratory i4HB—Institute for Health and Bioeconomy, Faculty of Pharmacy, University of Porto, 4050-313 Porto, Portugal; rsilva@ff.up.pt; 2UCIBIO—Applied Molecular Biosciences Unit, Laboratory of Toxicology, Department of Biological Sciences, Faculty of Pharmacy, Porto University, 4050-313 Porto, Portugal; 3Associate Laboratory i4HB—Institute for Health and Bioeconomy, University Institute of Health Sciences—CESPU, 4585-116 Gandra, Portugal; ana.sobral@iucs.cespu.pt (A.F.S.); ricardo.dinis@iucs.cespu.pt (R.J.D.-O.); 4UCIBIO—Applied Molecular Biosciences Unit, Toxicologic Pathology Research Laboratory, University Institute of Health Sciences (1H-TOXRUN, IUCS-CESPU), 4585-116 Gandra, Portugal; 5UCIBIO—Applied Molecular Biosciences Unit, Translational Toxicology Research Laboratory, University Institute of Health Sciences (1H-TOXRUN, IUCS-CESPU), 4585-116 Gandra, Portugal; 6Department of Public Health and Forensic Sciences and Medical Education, Faculty of Medicine, University of Porto, 4200-319 Porto, Portugal; 7FOREN—Forensic Science Experts, Dr. Mário Moutinho Avenue, No. 33-A, 1400-136 Lisbon, Portugal

**Keywords:** paraquat, myelin, oligodendrocytes, Schwann cells, demyelination, neurotoxicology, public health

## Abstract

Paraquat (1,1′-dimethyl-4,4′-bipyridilium dichloride), a widely used bipyridinium herbicide, is known for inducing oxidative stress, leading to extensive cellular toxicity, particularly in the lungs, liver, kidneys, and central nervous system (CNS), and is implicated in fatal poisonings. Due to its biochemical similarities with the neurotoxin 1-methyl-4-phenylpyridinium (MPP+), paraquat has been used as a Parkinson’s disease model, although its broader neurotoxic effects suggest the participation of multiple mechanisms. Demyelinating diseases are conditions characterized by damage to the myelin sheath of neurons. They affect the CNS and peripheral nervous system (PNS), resulting in diverse clinical manifestations. In recent years, growing concerns have emerged about the impact of chronic, low-level exposure to herbicides on human health, particularly due to agricultural runoff contaminating drinking water sources and their presence in food. Studies indicate that paraquat may significantly impact myelinating cells, myelin-related gene expression, myelin structure, and cause neuroinflammation, potentially contributing to demyelination. Therefore, demyelination may represent another mechanism of neurotoxicity associated with paraquat, which requires further investigation. This manuscript reviews the potential association between paraquat and demyelination. Understanding this link is crucial for enhancing strategies to minimize exposure and preserve public health.

## 1. Introduction

Pesticides, particularly herbicides, are widely used chemicals commonly applied in agriculture to protect crops from pests. Herbicides can be grouped into distinct classes according to their mechanisms of action. Thus, depending on the class, herbicides may induce a specific set of adverse effects in humans, especially in cases of acute poisoning [1].

Paraquat (1,1′-dimethyl-4,4′-bipyridilium dichloride) is a non-selective contact herbicide of the bipyridinium chemical family (Figure 1A), classified as a photosystem I electron diverter and widely used worldwide. Paraquat’s toxicity primarily results from its ability to induce oxidative stress. Upon entering biological systems, paraquat undergoes redox cycling, generating reactive oxygen species (ROS) such as superoxide radicals (O_2_^•−^). These ROS cause extensive cellular toxicity through lipid peroxidation, protein oxidation, and DNA damage [2,3,4]. The lungs are particularly susceptible to paraquat’s harmful effects due to its accumulation in alveolar cells, leading to pulmonary fibrosis and respiratory failure [5,6]. Additionally, paraquat can affect other organs, including the liver, kidneys, and central nervous system (CNS), contributing to its overall systemic toxicity [6].

Due to the molecular and biochemical similarities of paraquat and 1-methyl-4-phenylpyridinium (MPP^+^; also known as cyperquat), the active form of the neurotoxin 1-methyl-4-phenyl-1,2,3,6-tetrahydropyridine (MPTP; Figure 2), which causes Parkinson-like symptoms in both animals and humans, paraquat has been increasingly investigated as a model of Parkinson’s disease [8,9]. However, as seen for several neurotoxicants, the effects of paraquat on the nervous system may result from a multiplicity of targets and mechanisms [2,4,8,9]. Thus, the impact of paraquat on the nervous system is expected to be profound.

Demyelinating diseases represent a large group of pathologies characterized by defects in myelin sheaths that surround neuronal axons. Among this group of diseases are multiple sclerosis (MS) (the most prevalent condition), neuromyelitis optica, acute disseminated encephalomyelitis, transverse myelitis, Guillain–Barré syndrome, chronic inflammatory demyelinating polyneuropathy, and central pontine myelinolysis (CPM), among others. These diseases can affect neurons in the CNS and peripheral nervous system (PNS), resulting in a wide spectrum of clinical manifestations with variable ages of onset, progression, and recovery [10,11,12,13,14]. The exact etiology of these pathologies remains elusive, although genetic predisposition, autoimmune mechanisms, and environmental factors are believed to play pivotal roles.

In recent years, growing concerns have emerged regarding the impact of chronic, low-level exposure to herbicides on human health. This issue is particularly serious due to agricultural runoff, which contaminates drinking water sources. Moreover, the presence of these chemicals in food supplies may have significant effects on human health. While the connection between demyelination and the development of demyelinating diseases has been recognized for some herbicides, including glyphosate (5-enolpyruvylshikimate-3-phosphate synthase inhibitor) [15] and diquat (Figure 1B; belongs to the same chemical family as paraquat) [16,17], it remains less evident for paraquat, despite its other well-known effects on the nervous system.

This manuscript reviews the available information linking paraquat and demyelination. Understanding this putative link could lead to more effective public health policies and regulations, as well as help developing targeted strategies to minimize exposure and associated risks. This knowledge could have broader implications for understanding the neurobiology of demyelinating diseases and may also inform research on other neurological diseases and conditions.

## 2. Herbicides

Herbicides are chemical substances used to control unwanted vegetation, showing varying levels of specificity. They play a crucial role in agriculture, gardening, and forestry by managing the growth of weeds that compete with crops for nutrients, water, and sunlight. Therefore, the use of herbicides has significantly increased agricultural productivity and efficiency. Globally, herbicides are estimated to make up nearly 48% of all pesticides used [18].

Chemically, these substances exhibit a wide range of structures and modes of action, some specific to plants and others shared with human biology. This raises concerns about environmental impact and human health. Most health issues in humans or animals from herbicide exposure result from improper use or the inaccurate disposal of containers, as such incidents are rare when the products are used correctly. However, growing concerns are emerging about the impact of herbicides on human health, particularly due to agricultural runoff contaminating drinking water sources and their presence in food [18,19].

### 2.1. Brief History of Herbicides

The use of substances to control weeds dates back thousands of years when early civilizations utilized primitive methods such as hand-pulling and crop rotation, as well as organic compounds like salt, ashes, and animal manure, to suppress undesired plants. These natural herbicides, although effective, lacked the precision and efficiency of modern chemical herbicides [20].

The modern era of herbicides began in the 19^th^ century during the Industrial Revolution, which provided advances in chemistry and agricultural science. The elucidation of the selective herbicidal effects of copper sulfate against *Sinapis arvensis* in 1895–1897 advanced the utilization of other natural compounds as herbicides, including sulfuric acid, sodium chlorate, borate, arsenite, arsenic trioxide, and dinitro-ortho-cresol [21]. However, these compounds lacked selectivity, which remained a considerable problem. This need resulted in an increased interest in developing selective herbicides to improve agricultural practices.

The first synthetic herbicides emerged in the early 20^th^ century, particularly in the 1940s, marking a significant advancement in the development of selective herbicides targeting specific types of plants without damaging crops. The most notable of these early selective herbicides was 2,4-dichlorophenoxyacetic acid (2,4-D), discovered by a team led by Dr. Franklin D. Jones in the United States. 2,4-D was innovative because it selectively targeted broadleaf weeds while leaving grasses, such as wheat and corn, intact [22,23].

Following the success of 2,4-D, the mid-20^th^ century was marked by a rapid evolution in herbicide development with the discovery of new classes of herbicides with distinct mechanisms of action. Triazines, such as atrazine, which was registered in 1958 by Ciba-Geigy for weed control in corn, became highly popular for controlling broadleaf and grassy weeds in crops like corn and sorghum [24]. Extensive use of atrazine led to widespread contamination of surface water, groundwater, and drinking water with this herbicide and its breakdown products [25]. Due to this contamination, European countries have increasingly restricted or prohibited the use of atrazine over the past decades. In contrast, the United States continued to extensively use atrazine despite these concerns [26,27,28]. However, these issues put more pressure on the development of new, safer herbicides.

Paraquat (Figure 1A) was initially identified as a chemical redox indicator by Weidel and Russo in 1882. Later, in 1933, Michaelis and Hill recognized its redox characteristics and named it methyl viologen. It was subsequently introduced as an herbicide for commercial use in 1962 [19]. A detailed exploration of paraquat as an herbicide is provided in Section 2.2.

Glyphosate (Figure 3) was patented by the Stauffer Chemical Co. in 1961 as a descaling and metal-chelating agent. In 1970, Monsanto chemist John E. Franz discovered glyphosate’s effectiveness in weed control, leading the company to market glyphosate under the brand name Roundup [29]. Due to its widespread use, Monsanto Company developed genetically modified crops resistant to glyphosate, allowing farmers to apply it without harming their cultures [30].

In recent years, we have been facing a reduction in herbicide use due to the adoption of precision agriculture technologies like auto-guidance, variable-rate application, and section control. This tendency demonstrates the need for agricultural innovation, particularly emphasizing the necessity of balancing productivity and sustainability [31].

### 2.2. An Overview of the Herbicide Paraquat

Paraquat (1,1′-dimethyl-4,4′-bipyridilium dichloride) is a non-selective contact herbicide from the class of bipyridyl derivatives, known for its rapid and effective control of a wide variety of weeds and grasses in farms orchards, as well as general weed management.

Paraquat is currently restricted or banned in over 60 countries due to its acute and long-term toxicity. The European Union banned paraquat in 2007, citing severe health risks, including fatal poisonings, with similar actions taken by countries across Asia, Africa, and South America [32,33]. However, paraquat continues to be used on farms in the United States, where it remains widely applied despite strict regulations that require special licensing and training for applicators to mitigate its risks [34]. Other countries, such as India, have minimal or no regulations restricting its use [35].

Paraquat’s effectiveness as an herbicide is due to its redox potential, involving a continuous cycle of reduction and reoxidation in the presence of O_2_, thereby generating high levels of ROS, including O_2_^•−^ and hydroxyl radicals (HO^•^) (Figure 4). However, this redox mechanism is also implicated in paraquat’s toxicity in mammals [36]. In fact, paraquat is considered one of the most toxic commonly used herbicides. For instance, paraquat has an oral LD_50_ in rats of 100 mg/kg [37] and an even lower LD_50_ in other species, including monkeys [38], dogs [39,40], guinea pigs [38], and humans [41].

Paraquat has minimal absorption from the skin but is rapidly absorbed from the gastrointestinal tract in both laboratory animals and humans. Distribution studies have shown that paraquat particularly accumulates in lung type I and type II epithelial cells and Clara cells, reaching concentrations 6- to 10-fold higher than in the bloodstream. Due to the high availability of O_2_, paraquat’s continuous redox cycling generates massive amounts of O_2_^•−^ that attacks polyunsaturated lipids, causing the peroxidation of membrane lipids. This leads to extensive pulmonary lesions and fibrosis, making the lungs the primary target for paraquat toxicity [6]. Apart from lung toxicity, paraquat also accumulates in the kidneys, making them sensitive to herbicide toxicity [42]. In addition, paraquat is transported to the brain by the neural amino acid transporter. This accumulation in the brain has been implicated as a risk factor for developing Parkinson’s disease [9].

Paraquat undergoes minimal metabolism in the human body and is mainly eliminated unchanged in urine and feces (the amount of paraquat excreted in feces corresponds to 60–70% of the ingested dose) [43,44].

Previous studies in rodents provided evidence of teratogenic effects consequent to paraquat exposure. The administration of high doses of paraquat to mice (1.67 and 3.35 mg/kg i.p. or 20 mg/kg *per os*, daily) on days 8–16 of gestation increased the nonossification of the sternebrae [45]. Additionally, the administration of paraquat to pregnant mice (100 ppm in drinking water, starting on day 8 of gestation and continuing the exposure of the newborns for 42 days after birth) increased the postnatal mortality of their offspring [46]. Supporting these findings, research on developing *Xenopus laevis* embryos has shown that paraquat can cause developmental abnormalities, including ventral tail bending, notochord curvature, and stunted growth, all indicative of teratogenic effects [47,48,49]. Overall, these data suggest a teratogenic potential for paraquat.

Since its introduction as an herbicide in 1962, paraquat has been implicated in several human intoxications, many of them fatal (232 human deaths between 1964 and 1973) [50]. In fact, acute exposure to low amounts of paraquat can be fatal within a few days. Most of these cases involved the intentional ingestion of concentrated formulations for suicide purposes. However, other cases involved accidental ingestion from improperly labeled bottles. This prompted manufacturers to include distinct features in the paraquat formulations, including a blue pigment and a compound with an intense odor, to reduce the likelihood of accidental ingestion [36].

In addition to acute toxicity, chronic exposure to low levels of paraquat has been considered a predisposing factor for developing Parkinson’s disease [9].

### 2.3. Mechanisms of Paraquat Toxicity

The mechanisms through which paraquat exerts its toxic effects in humans mainly involve an interplay between oxidative stress, mitochondrial dysfunction, and the activation of inflammatory pathways.

Critical to its toxicity is paraquat’s ability to undergo redox cycling. By accepting an electron from cellular nicotinamide adenine dinucleotide phosphate (NADPH), a paraquat divalent cation (PQ^2+^) can be easily converted to its monovalent cation free radical (PQ^•+^) (Figure 5). This radical reacts with molecular oxygen (O_2_), generating O_2_^•−^ and regenerating PQ^2+^ [6]. The excessive production of ROS, coupled with the depletion of the cell’s natural antioxidant defenses—particularly NADPH and, consequently, reduced glutathione (GSH)—leads to oxidative damage to vital macromolecules, including lipids (lipid peroxidation), proteins (protein carbonylation), and nucleic acids (DNA oxidation) [51,52,53,54]. In cell types with inherently weaker antioxidant defenses, such as neurons, oxidative stress can result in significant cellular damage and lead to cell death [55] (Figure 5). Moreover, when excessive oxidative stress compromises the plasma membrane integrity, unregulated cell death (necrosis) may occur [51]. Necrosis often results in additional inflammation and widespread tissue damage, exacerbating the harmful effects of paraquat exposure [56].

In mammals, paraquat also enters the mitochondria and disrupts their function by interfering with the electron transport chain (Figure 5). This interference impairs the production of adenosine-5′-triphosphate (ATP) and exacerbates ROS generation [2,57]. The resultant mitochondrial dysfunction manifests as alterations in mitochondrial membrane potential and permeability, leading to the release of pro-apoptotic factors such as cytochrome c and the activation of caspases, which trigger programmed cell death through apoptosis [58,59].

Paraquat also triggers systemic inflammatory responses (Figure 5). The oxidative stress and cellular damage induced by paraquat activate immune cells, leading to the release of pro-inflammatory cytokines and the initiation of inflammatory signaling pathways [59,60]. Key pathways affected include the activation of the inflammatory transcription factor nuclear factor kappa B (NF-κB) and the NOD-, LRR- and pyrin domain-containing protein 3 (NLRP3) inflammasome, a multiprotein complex that drives pro-inflammatory and apoptotic cascades, exacerbating tissue damage [60,61,62,63]. Additionally, other signaling pathways involved in regulating inflammation, such as mitogen-activated protein kinases (MAPKs), are also impacted by paraquat exposure [60,64]. With long-term exposure, this chronic inflammation can affect the function of multiple organs.

## 3. Biological Processes Underlying Myelination

Myelination refers to the formation of the myelin sheath that protects the axons and facilitates the efficient propagation of electric signals in the nervous system. Therefore, understanding the biological processes underlying myelination is crucial for comprehending the demyelination process and for developing therapeutic strategies aiming at improving patient outcomes.

### 3.1. Myelin Structure and Function

Myelin is a specialized, multilayered, lipid-rich sheath composed of 40 or more closely packed layers of a lipid bilayer that envelops the axons of neurons [65]. The exact time required for myelin renewal in humans remains uncharacterized. However, studies in laboratory animals have suggested that the myelin sheath undergoes continuous remodeling, with distinct turnover rates throughout life [66]. Thus, the biosynthesis, storage, and transport of myelin components are essential to guarantee proper myelin remodeling during the entire life [67].

Myelin represents a structure with a low water content of approximately 40%, compared to the 80% water content characteristic of gray matter. By dry weight, myelin is primarily composed of a high content of lipids (about 70–85%) and a low amount of proteins (about 15–30%) [68]. By contrast, conventional cell membranes generally comprise about 50% of lipids and 50% of proteins [69]. This characteristic of myelin allows it to form an effective insulating layer. The lipid content of myelin includes phospholipids, cholesterol, and glycolipids, all of which are critical for the insulating properties of the myelin sheath [70,71,72].

The main functions of myelin are to speed up the propagation of action potentials and to support metabolic coupling. In myelinated axons, the action potential exhibits a saltatory transmission, which provides faster and more efficient propagation of electric signals [73]. This is achieved because the myelin sheath is securely anchored to the axon at each end of an internode, preventing any leakage below the myelin sheath. In these regions, the myelin membrane does not form a compact structure but instead terminates in paranodal loops [74].

### 3.2. Myelination by Oligodendrocytes and Schwann Cells

Oligodendrocytes are the myelinating cells responsible for axon myelination in the CNS, while Schwann cells myelinate axons in the PNS. Thus, although oligodendrocytes and Schwann cells are thought to have a common precursor and lineage [75], they differ substantially in their targets. While oligodendrocytes are capable of myelinating axons from distinct neurons, Schwann cells can only form myelin sheaths on one axonal segment of a single neuron. Another significant aspect is related to the completion of myelination between the CNS and PNS. While in the PNS most myelination is completed shortly after birth, in the CNS, myelination is a progressive process that continues throughout adulthood [76,77]. Such differences may explain why demyelinating diseases are typically confined to those affecting either PNS myelinated fibers, such as Charcot–Marie–Tooth disease type 1 (CMT1), or CNS axons, such as MS and leukodystrophies.

## 4. Pathophysiology of Myelin Loss

Demyelinating diseases are a group of disorders characterized by damage to the myelin sheath that surrounds nerve fibers in the CNS and PNS. They can be categorized into two distinct groups: conditions that affect the CNS and those that affect the PNS. CNS demyelinating diseases result from the loss of myelin in CNS neurons. Based on their pathogenesis, they can be categorized into several groups: demyelination due to inflammatory processes (such as MS, neuromyelitis optica, transverse myelitis, acute disseminated encephalomyelitis, and acute hemorrhagic leukoencephalitis), viral demyelination (e.g., progressive multifocal leukoencephalopathy), demyelination associated with acquired metabolic disorders (like central pontine and extrapontine demyelination, adrenoleukodystrophy/adrenomyeloneuropathy), hypoxic-ischemic forms of demyelination, and demyelination caused by focal compression [10,11,12].

On the other hand, PNS demyelinating diseases affect PNS neurons and represent a large group of conditions that, based on their pathogenesis, can be categorized into two distinct categories: primary, inherited conditions, and those that are acquired. Primary peripheral demyelinating neuropathies with inherited transmission encompass Charcot–Marie–Tooth disease type 1 (CMT1), Dejerine–Sottas syndrome, congenital hypomyelinating neuropathy, and hereditary neuropathy with liability to pressure palsies. By contrast, Guillain–Barré syndrome, anti-MAG neuropathy, chronic inflammatory demyelinating polyradiculoneuropathy, and POEMS syndrome (POEMS term arises from the following signs and symptoms: Polyneuropathy, Organomegaly, Endocrinopathy/edema, Monoclonal-protein, and Skin changes) are collectively considered acquired peripheral demyelinating neuropathies [11,13].

Diverse mechanisms are involved in the pathophysiology of these conditions. They include immune-mediated mechanisms, genetic and environmental factors, damage of myelinating cells, dysfunction of astrocytes, or microglia activation (Figure 6). In demyelinating diseases like MS, the immune system plays a significant role in the pathophysiology of the disease by incorrectly recognizing myelin as an external component. This results in the release of cytokines and other inflammatory mediators that contribute to tissue damage. Consequently, the neuroinflammation characteristic of the disease can damage myelin and myelinating cells [78].

Variants of the HLA-DRB1 gene, particularly HLA-DRB1*15:01 [79], and distinct IL7RA loci (rs3194051, rs987107, and rs11567686) [80] are strongly associated with an increased risk of MS. Mutations in the gene coding for peripheral myelin protein 22 (PMP22) are associated with CMT1 [81]. On the other hand, previous infection with the Epstein–Barr virus, infectious mononucleosis, or cigarette smoking are recognized as environmental susceptibility factors for MS [82].

Astrocytes play a crucial role in myelination and maintaining CNS stability. Any disruption in their function can impair remyelination and worsen demyelinating diseases [83]. For instance, reactive astrocytes are characteristic of MS, which release cytokines and chemokines that inhibit the function of oligodendrocyte progenitor cells (OPCs) and myelination [84,85].

Additionally, activated microglia are the primary drivers of demyelination and oligodendrocyte loss, with the depletion of microglia significantly reducing demyelination and associated pathology [86]. Thus, demyelination can result from a complex interplay between immune-mediated, genetic, and environmental factors, along with the involvement of astrocytes and microglia, which disrupts myelin integrity in both the CNS and PNS.

## 5. Pathophysiology of Chemical-Induced Demyelination

Chemicals can induce demyelination through a multitude of pathways, reflecting their diverse chemical properties and modes of action [87]. Myelin loss can occur due to a direct attack on the myelin sheath or as a result of the disruption or death of myelinating cells (oligodendrocytes or Schwann cells). The distinction between these two different causes of demyelination may not always be easily identifiable, as they can sometimes occur simultaneously. Despite this variability, the ultimate outcome of demyelination is consistent [88]. Other mechanisms can also lead to myelin loss, including inflammatory responses and immune system activation, metabolic alterations (such as deficiencies in vitamins or minerals), oxidative stress and mitochondrial dysfunction, traumatic injury, and neoplastic conditions, among others [88]. In conditions like MS, immune-mediated mechanisms are prominent [89]. Cadmium has been shown to induce mitochondria-dependent apoptosis in oligodendrocytes [90]. Lead diffuses and accumulates in the brain, impairing the formation of myelin components, thereby causing hypomyelination and demyelination [91]. The environmental toxin bisphenol A (BPA) has been linked to the degeneration of immature and mature oligodendrocytes, resulting in altered expression patterns of myelin-related genes [92]. Overall, common chemicals induce a multiplicity of defects that result in defective myelination or demyelination.

## 6. Does Paraquat Cause Demyelination?

Although several studies have addressed the general toxicity and carcinogenic potential of herbicides, fewer studies have explored their influence on the nervous system. However, this scientific elucidation is a topic of continuous interest due to the characteristic neurotoxic effects observed upon exposure to some herbicides and their widespread use. Additionally, while the potential impact of herbicides on axonal myelination, an essential mechanism for ensuring proper neuronal function and health, has only been sporadically and superficially explored, it represents a topic of utmost importance.

### 6.1. The Relationship Between Paraquat and Demyelination

The influence of paraquat on myelination and, consequently, its involvement in demyelinating conditions, remains poorly explored. Nevertheless, some studies have reported that paraquat has the potential to exert direct toxic effects on myelinating cells, influence the expression of myelin-related genes, alter myelin structure, and induce neuroinflammation [62,93,94,95,96,97,98] (Figure 7). This raises the hypothesis that paraquat may cause demyelination.

#### 6.1.1. Direct Toxic Effects of Paraquat on Myelinating Cells

Early studies from 2004 by Ernst and colleagues [96] provided direct evidence that paraquat is toxic to oligodendrocytes in vitro. The authors used the oligodendrocyte cell line OLN 93, which expresses galactocerebroside and myelin-related proteins, including myelin basic protein (MBP), myelin-associated glycoprotein (MAG), proteolipidprotein (PLP), and Wolfgram protein (WP), but does not exhibit astrocytic properties [99]. They showed a concentration-dependent cell death following paraquat exposure (up to 1000 µM, for 48 h). These effects were accompanied by significant increases in protein carbonyl content (387 ± 42 pmol/mg protein for control versus 642 ± 66 pmol/mg protein after 125 µM of paraquat) [96]. These results support the idea that paraquat may exert direct toxic effects on myelinating cells in vitro and, therefore, it may have the potential to induce myelin loss.

#### 6.1.2. Effects of Paraquat on Myelin-Related Genes and Myelin Structure

Apolipoprotein D (ApoD) is a lipocalin lipid-binding protein expressed in oligodendrocytes [100], Schwann cells [101], pericytes [102], and astrocytes [103] and possesses antioxidant properties. Although mature neurons do not express ApoD, they are capable of internalizing this protein from the surrounding extracellular environment, especially under conditions of oxidative stress [104]. Thus, altered ApoD expression and/or function are likely to impact both the producing and the target cells, including neurons.

Studies in cultured cells and animal models have shown that ApoD is essential for nervous system homeostasis and normal function, as well as for the development and preservation of crucial neural structures [105]. Specially, ApoD is involved in various myelin-related functions, including myelin maintenance, myelin clearance following axonal injury, and the negative regulation of macrophage recruitment to damaged axonal areas [106]. Moreover, in the absence of ApoD, axon regeneration and remyelination are delayed, and reduced myelin sheath thickness is detected even in uninjured nerves [106].

In 2011, while investigating the effect of ApoD on the early transcriptional changes upon oxidative stress, Bajo-Grañeras and colleagues [97] revealed a downregulation of myelin-related genes in the cerebellum of wildtype C57BL/6J mice administered with paraquat (30 mg/kg), as evaluated 6 h after herbicide administration. Particularly, they found a downregulation of *Ugt8a* (fold-change = −2.17), *Sox2* (fold-change = −2.29), *Sox10* (fold-change = −2.41), *Cldn5* (fold-change = −2.44), *Erbb3* (fold-change = −2.55) and *Nr4a2* (fold-change = −2.69) genes [97]. *Ugt8a* gene encodes for UDP Galactosyltransferase 8A, an enzyme essential for myelin maintenance [107,108]. *Sox2* gene encodes SRY-box-containing gene 2, a critical transcription factor essential for oligodendroglia proliferation and differentiation during postnatal brain myelination and CNS remyelination [109]. *Sox10* gene encodes SRY-box-containing gene 10, which is also a transcription factor essential for myelin maintenance and gene expression. Heterozygous loss-of-function mutations in this gene have been implicated in peripheral demyelinating neuropathy and central demyelination [110]. *Cldn5* gene codes for Claudin 5, which plays a major role in tight junction-specific obliteration of the intercellular space at the BBB, thus impacting myelin integrity [111]. *Erbb3* gene encodes for V-erb-b2 erythroblastic leukemia viral oncogene homolog 3, which is essential for Schwann cell migration and myelination [112]. Lastly, the Nr4a2 gene-encoding product, nuclear receptor subfamily 4, group A, member 2 (NURR1), is upregulated in MS patients in the pre-disease state [113] and prevents autoimmune neuroinflammation in the experimental autoimmune encephalomyelitis (EAE) model of MS [114]. Notably, a significant downregulation of *Ugt8a*, *Sox2*, *Sox10* and Cldn5 gene expression was not detected in the cerebellum of ApoD knockout mice administered with paraquat (fold-change for wildtype: −2.25, −2.28, −2.31 and −2.41, respectively, compared to the cerebellum of ApoD knockout mice) [97]. These results indicated that paraquat interferes with myelin-related genes and that these effects depend on ApoD. Supporting a role for ApoD in paraquat-induced alterations in myelin-related genes, Diez-Hermano and colleagues [115] observed a marked upregulation of ApoD variant E in the cerebellum of 6-month-old C57BL/6 mice six hours after administering paraquat (30 mg/kg, i.p.).

Effective lysosomal function is also essential for myelination by Schwann cells and oligodendrocytes [116,117], and lysosomal dysfunction has been increasingly recognized as a fundamental factor in the development of neurodegenerative diseases [118,119]. ApoD has also been shown to prevent lysosomal alkalinization by paraquat, thus protecting against paraquat-induced lysosomal damage in astrocytes and neurons [117]. Collectively, these findings provide strong evidence that paraquat-mediated interference with ApoD dynamics could also significantly disrupt lysosomal function, potentially triggering a cascade of events that could lead to demyelination.

Overall, these data suggest that paraquat could significantly impact myelin structure and disrupt lysosomal function, potentially predisposing individuals to demyelination.

An in vivo study performed by Hichor and colleagues in eight-week-old male C57BL/6 mice also provided direct evidence on the impact of paraquat on myelin structure [98]. Animals administered with a single i.p. dose of paraquat (30 mg/kg) showed thicker myelin sheaths around fibers of the sciatic nerve (axon perimeter/nerve fiber perimeter: control = 0.7055 ± 0.0030; paraquat = 0.6206 ± 0.0056), as evaluated 1 week after paraquat administration [98]. This was indicative of altered myelin structure. Furthermore, these paraquat-induced alterations in myelin structure were accompanied by a 3-fold increase in 2-hydroxyethidium formation (an indicator of O_2_^•−^ production), a 2-fold increase in malondialdehyde formation (indicator of lipid peroxidation), a 2.5-fold increase in protein carbonyls (indicator of protein oxidative modifications), and a decreased ratio of reduced glutathione (GSH)/oxidized glutathione (GSSG) in the sciatic nerves of mice. This is likely a reflex of paraquat’s capability to generate oxidative stress. In addition, the sciatic nerves of paraquat-administered mice showed reduced expression, at both the protein and mRNA levels, of the myelin components PMP22 and myelin protein zero (MPZ), as detected 48 h and 1 week post-paraquat administration. Mechanistically, the alterations in myelin gene expression caused by paraquat were linked to the activation of the liver X receptor (LXR) pathway, which subsequently inhibited the Wnt/β-catenin signaling pathway, a well-established regulator of myelin gene expression [98]. Based on these results, it can be hypothesized that paraquat may induce peripheral neuropathy. In line with this hypothesis, paraquat-administered animals showed multiple signs of peripheral demyelination. These include the following (compared to control): (1) a reduced number of central and peripheral square crossings (indicative of less exploration) in the open field test, as evaluated 48 h and 1 week after paraquat administration; (2) movements 2.5-fold slower, twice as many stops, and three-times more hind-paw slips, as measured at the 48 h timepoint; (3) a significant decrease in limb muscular strength (48 h and 1 week after paraquat administration); and (4) a reduced leg withdrawal reflex, as evaluated by the hot-plate test 1 week after paraquat administration [98]. Overall, this work provided the first direct evidence that paraquat targets myelin components and induces peripheral demyelination.

#### 6.1.3. Paraquat and Neuroinflammation

In vitro studies using microglia BV2 cells have demonstrated paraquat’s ability to trigger an inflammatory response [62,94,95]. The morphological alterations of BV2 cells (round bodies and shortened protuberances) induced by paraquat (20, 40, or 80 µM for 6, 12, or 24 h) were associated with a time- and concentration-dependent increase in the levels of pro-inflammatory cytokines, including TNF-α, interleukin-1β (IL-1β), and interleukin-6 (IL-6), detected at both the protein and mRNA levels [94]. Even at non-cytotoxic concentrations, paraquat (0.015, 0.03, 0.06, and 0.12 µM for 24 h) was capable of increasing the levels of these pro-inflammatory cytokines, inducing microglial cells toward an M1 phenotype, and enhancing their migration and phagocytic activity [62].

A similar activation of BV2 microglia induced by paraquat (0.12 and 1 µM for 24 h) was later reported by another study [95]. The exposure of primary cortical neurons to supernatants of paraquat-induced activated BV2 microglia resulted in a significant reduction of neuronal viability. Additionally, the exposure of PC12 cells to supernatants of paraquat-induced activated BV2 microglia reduced cell proliferation and significantly increased the levels of the pro-inflammatory cytokines TNF-α and IL-6 in a concentration-dependent manner [95]. Mechanistically, this pro-inflammatory condition was caused by the activation of the TLR4 signaling cascade and the subsequent activation of the NF-κB signaling pathway [62,94,95].

In line with these observations, Imam and colleagues [93] recently showed that the administration of paraquat to mice (10 mg/kg i.p., once daily for 14 days) also resulted in significant neuroinflammation in the substantia nigra pars compacta and cerebellum. This was evidenced by a marked increase in the pro-inflammatory cytokines TNF-α, interleukin-1 (IL-1), and IL-6, alongside a notable decrease in the anti-inflammatory cytokine interleukin-10 (IL-10) in these brain regions of paraquat-administered mice. Additionally, paraquat administration resulted in the decreased staining of most myelin fibers and a significant reduction in the immunostaining of key oligodendrocyte markers, including platelet-derived growth factor receptor A (PDGFR-α) and oligodendrocyte transcription factor 2 (Olig-2) [93]. Olig-2 is a transcription factor that stimulates the expression of myelin-associated genes [120].

Based on these findings, it can be concluded that paraquat significantly contributes to neuroinflammation by inducing pro-inflammatory cytokine production, activating microglial cells, and impairing both myelin integrity and oligodendrocyte function. Notably, neuroinflammation and demyelination are interconnected processes often observed in various neurological disorders, including certain demyelinating diseases [121]. These results strongly support the hypothesis that paraquat may induce demyelination.

## 7. Impact on Human Health

Paraquat is still one of the most widely used herbicides globally. Although still limited, some evidence supports a putative effect of paraquat on myelination. In line with this, the consequences for human health are remarkably significant, including cognitive and motor impairments, reducing an individual’s quality of life and resulting in a substantial socioeconomic burden for society.

Several studies have associated paraquat with cognitive and motor impairments [122,123,124,125]. In the context of paraquat exposure, cognitive impairment may also arise as a consequence of other factors. The inflammatory cytokines released during neuroinflammation can affect neurotransmitter systems. Dopamine is a known regulator of cognitive function [126], and an impairment of astrocyte-dependent dopamine homeostasis in the developing prefrontal cortex results in cognitive deficits [127]. Dopamine function is particularly vulnerable to elevated levels of inflammatory cytokines [128,129]. Paraquat has been shown to trigger an elevation of inflammatory cytokines in microglia [62,94,95], and PQ-induced inflammatory microglia further induce secondary inflammation in neurons [95]. Therefore, it is likely that a paraquat-induced elevation of inflammatory cytokines may also perturb dopamine homeostasis, resulting in cognitive deficits.

Paraquat is also a known substrate for the dopamine transporter (DAT). While PQ^2+^ is not transported by DAT, it can be easily converted to PQ^•+^ by the microglial NADPH oxidase, making it a DAT substrate. Its accumulation in DAT-containing neurons induces oxidative stress and cell death [130]. As a DAT substrate, paraquat may also perturb dopamine neurotransmission. Supporting this, paraquat has been shown to disrupt the homeostasis of the dopaminergic system in PC12 cells [131] and is extensively used as a model for Parkinson’s disease due to its ability to cause a degeneration of dopaminergic neurons [8,9].

Glutamate also plays an important role in cognition, with research concentrated on the pharmacological manipulation of glutamatergic neurotransmission to enhance cognitive function [132]. At this level, the pro-inflammatory cytokine IL-1β, whose expression is increased by paraquat-induced activation of an inflammatory response in BV2 cells [62,94], has been shown to inhibit synaptic transmission mediated by glutamate-activated *N*-methyl-*D*-aspartate (NMDA) receptors [133]. IL-1β may also interfere with astrocyte-mediated glutamate uptake [134]. IL-6 and TNF-α, which are both upregulated in paraquat-activated microglial BV2 cells [62,94,95], also modulate glutamatergic neurotransmission. For instance, IL-6 inhibits the release of glutamate in the cerebral cortex [135]. In addition, TNF-α increases presynaptic glutamate release in cultured neurons [136] and inhibits the activity of glutamate transporters [137]. Thus, it is likely that a paraquat-induced elevation of inflammatory cytokines may also perturb glutamatergic neurotransmission, resulting in cognitive deficits.

### 7.1. The Relationship Between Demyelination and Cognitive and Motor Impairments

The relationship between demyelination and cognitive impairment is a growing area of research. A likely consequence of demyelination is the disruption of the integrity of neural circuits and the impairment of communication between different brain regions, leading to cognitive deficits. Studies have shown that individuals with demyelinating diseases often exhibit difficulties in memory, attention, and executive function [138,139,140,141]. Studies in the cuprizone animal model of demyelination have shown that demyelination leads to long-term cognitive impairments characterized by deficits in spatial working memory, social memory, and cognitive flexibility, as well as increased hyperactivity, even after remyelination [140]. More strikingly, in cognitively unimpaired individuals, reduced myelin content is linked to a faster cognitive decline [141].

On the other hand, the structural integrity of the brain may be compromised by the loss of oligodendrocytes—the cells responsible for myelinating CNS neurons. When oligodendrocytes are damaged or lost, the resulting demyelination disrupts neural communication, exacerbating the effects of demyelination on cognitive function. This is supported by the observations that mice administered with cuprizone, a toxin that induces oligodendrocyte apoptosis, show cognitive impairment [142]. Moreover, oligodendrocyte loss is associated with several demyelinating conditions, including MS and certain types of leukodystrophies [143], which often manifest in impaired memory, attention, and executive function [138,139,140,141]. Additionally, studies have shown that preserving oligodendrocyte function and promoting remyelination can be crucial in mitigating cognitive decline and improving overall neurological health [144].

A loss of myelin integrity also contributes to motor impairments. The demyelination of different CNS areas can impair the brain’s ability to initiate and coordinate movement [145,146,147]. In the PNS, demyelination can affect the nerves that transport signals from the spinal cord to muscles, leading to weakness and reduced motor function [13].

The fact that paraquat induces direct toxic effects on myelinating cells, changes the expression of myelin-related genes, and alters myelin structure suggests that its effects on cognitive and motor function might be related, at least to some extent, to the potential demyelination induced by the herbicide. However, this association still lacks direct evidence. Therefore, further research is of utmost importance to establish a direct causal link between paraquat-induced demyelination and its impact on cognitive and motor functions.

### 7.2. Consequences of Cognitive and Motor Impairments on Individuals’ Quality of Life and Long-Term Prognosis

The cognitive and motor impairments resulting from paraquat exposure, whether related to demyelination or not, significantly affect the quality of life of affected individuals. Cognitive deficits, encompassing impairments in memory, attention, and executive function, can significantly challenge daily routines and reduce independence. Individuals may experience difficulties in performing routine tasks, managing finances, or maintaining social relationships. The onset of these cognitive impairments can be particularly evident in individuals who previously exhibited normal cognitive function but subsequently developed deficits following paraquat exposure.

Motor impairments, including weakness, tremors, and coordination difficulties, may represent additional challenges faced by paraquat-exposed individuals. The loss of fine motor skills can significantly hinder an individual’s capacity to engage in activities that were once enjoyable or essential, such as writing, typing, or participating in sports. The cumulative effects of cognitive and motor impairments can lead to increased dependence on caregivers and healthcare services, ultimately resulting in a diminished quality of life.

Additionally, the psychological impact of living with chronic impairments should not be underestimated. Studies have indicated that paraquat may be associated with the development of neuropsychiatric conditions, including anxiety or depression [122,148]. The combination of physical limitations and cognitive impairment can contribute to isolation and reduced social interactions, further affecting overall well-being [149].

On the other hand, studies using animal models have demonstrated that long-term cognitive impairments can persist even after remyelination [140]. Furthermore, the chronic inflammatory state induced by paraquat exposure may predispose individuals to developing neurodegenerative diseases. Although the link is not clear, even for Parkinson’s disease [150,151], further research is essential to decipher whether there is a direct causal link between paraquat and neurodegenerative diseases, including demyelinating conditions.

### 7.3. Consequences of Cognitive and Motor Impairments on Socioeconomic Burden

Any impairment resulting from paraquat exposure will have socioeconomic implications. Individuals affected by these impairments often experience a reduced quality of life. The inability to work or the need for specialized care can result in significant financial pressure on families and caregivers. Moreover, the healthcare costs associated with managing cognitive impairments are substantial. These costs include medical treatments, rehabilitation services, and long-term care.

The broader economic impact extends beyond direct healthcare costs. Cognitive and motor impairments can reduce workforce productivity and increase absenteeism. Individuals with these impairments may require disability benefits and social support services, enhancing the financial burden on social welfare systems. The cumulative effect of these factors contributes to a substantial socioeconomic burden.

## 8. Concluding Remarks

This manuscript highlights studies suggesting a hypothetical link between paraquat and demyelination, specifically through direct toxic effects on myelinating cells and its role in promoting neuroinflammation—a key factor in certain demyelinating diseases.

The interplay between neurotoxicants and demyelinating diseases represents a significant public health concern, warranting greater attention and preventive measures. By elucidating the mechanism(s) underlying neurotoxicant-induced demyelination and enhancing our understanding of the associated health risks, we can progress more effectively in developing both preventive and therapeutic strategies.

The mechanism of action of paraquat primarily involves the generation of ROS via redox cycling, leading to oxidative stress and damage to cellular structures. The limited evidence linking paraquat and demyelination shows that paraquat can disrupt the expression of crucial myelin-related genes, interfere with oligodendrocyte function, and compromise myelin integrity. This disruption is not limited to myelination, as paraquat can also induce neuroinflammation, characterized by the upregulation of pro-inflammatory cytokines such as TNF-α, IL-1β, and IL-6, which may exacerbate oxidative stress, causing cellular damage. The interplay between neuroinflammation and demyelination is particularly important, as it suggests that paraquat may directly damage myelin while also creating a neurotoxic environment that favors/promotes neuronal damage. However, we must interpret these data with caution to avoid drawing incorrect conclusions based on similar connections.

On the other hand, exposure to paraquat is increasingly linked to cognitive and motor impairment. Cognitive functions, especially those mediated by dopaminergic and glutamatergic systems, are vulnerable to the inflammatory environment activated by paraquat. Thus, the well-established relationship between demyelination, cognitive decline, and motor impairment suggests a potential link between paraquat, demyelination, and these impairments. However, for now, without more consistent and robust data, these associations should not be overstated. Therefore, further investigations on the long-term effects of paraquat exposure on both human populations and animal models could provide valuable insights into its role as a neurotoxicant, particularly regarding demyelination.

## 9. Future Perspectives

As the use of herbicides continues to represent a global problem, understanding the potential long-term impacts on the nervous system, particularly in relation to demyelination, becomes increasingly important. Despite some evidence linking paraquat to demyelination, the exact mechanism(s) remain poorly understood. Studies have suggested a significant relationship between lung health and brain function and pathology [152,153]. Lung function can influence brain processes, potentially due to the brain dependency on efficient blood oxygenation [154]. Thus, poor lung function, often associated with low oxygen levels and inflammation, is linked to cognitive decline and accelerates the progression to dementia [153,155]. Particularly, chronic respiratory diseases, such as chronic obstructive pulmonary disease and asthma, have been correlated with an increased risk of neuroinflammation, oxidative stress, and even Alzheimer’s disease [156,157,158]. Considering that lungs are especially vulnerable to the harmful effects of paraquat, which accumulates in alveolar cells, causing pulmonary fibrosis and eventually respiratory failure [5,6], it can be hypothesized that such lung damage might contribute to paraquat’s effects on the CNS, including on myelin and myelinating cells. However, this relationship remains unclear, and future research should aim to uncover the pathways involved in paraquat-induced toxicity in myelinating cells, including this putative relationship. At this level, single-cell and single-nucleus RNA sequencing [159], single-cell and single-nucleus assay for transposase-accessible chromatin (ATAC) sequencing [160], single-cell epigenomics [161], single-cell proteomics [162], or multiomics at the single-cell level [163] may provide valuable insights into changes within specific cell populations associated with paraquat exposure. These advanced techniques have been successfully employed to examine cell-specific phenotypes and disrupted cellular processes, as myelination, in neurodegenerative and neuroinflammatory disease models, including Alzheimer’s disease, Parkinson’s disease, and MS, holding significant promise for applications in neurotoxicology studies [159].

On the other hand, the available in vivo studies primarily focus on acute exposures to relatively high concentrations of paraquat [97,115,164]. This experimental approach may amplify the effects of paraquat, enabling a more precise elucidation of the underlying mechanistic pathways behind its toxic effects. Notably, exposure to a high dose of paraquat can induce Parkinson’s disease-like symptoms [4]. Thus, future research examining the substantia nigra (the key pathological brain region affected in Parkinson’s disease [165]) in paraquat-exposed animals could provide valuable insights into paraquat’s effects on myelin and myelinating cells.

Nevertheless, acute exposures to high concentrations of paraquat may not accurately represent the scenarios of prolonged exposure to low levels of the herbicide needed to evaluate its cumulative effects. Thus, longitudinal epidemiological studies are needed to analyze populations chronically exposed to low levels of paraquat for extended periods, focusing on cognitive and motor function assessments. Neurological examinations to identify signs of demyelination or other neurodegenerative conditions, particularly in agricultural communities where paraquat use is prevalent, would also be beneficial. Moreover, advancements in pathology and functional assessments will also be key in understanding the long-term impact of paraquat exposure on humans and animals in affected areas. Such knowledge is important for establishing any causal relationship between paraquat exposure and long-term effects on the CNS and PNS. Therefore, more studies are needed to elucidate the potential impact of paraquat on myelination and neuroinflammation over time.

Since food contamination with herbicides may contribute to chronic low-level exposure to these chemicals, improving strategies for detoxifying paraquat and other herbicidal contaminants in food is an important area of research. Several methods for reducing or eliminating herbicide residues in food have been explored. These strategies include (1) washing fruits and vegetables under running water to remove some surface residues of herbicides like paraquat [166]; (2) peeling fruits or vegetables with thicker skins, such as apples, potatoes, or cucumbers, to significantly reduce herbicide residue, and removing outer leaves of leafy vegetables where residues may accumulate [167]; (3) soaking in salt or vinegar solutions [166]; and (4) using cooking methods such as boiling, blanching, and stir-frying, which can break down certain herbicide residues [168]. These methods can help limit exposure to herbicides from contaminated food.

Another important aspect is co-exposure to multiple neurotoxic compounds, which represents a real scenario in contemporary society. It is plausible that the co-exposure to other neurotoxicants may exacerbate the putative effects of paraquat in myelination. Thus, understanding these interactions and their implications for human health is of utmost importance.

## Figures and Tables

**Figure 1 antioxidants-13-01354-f001:**
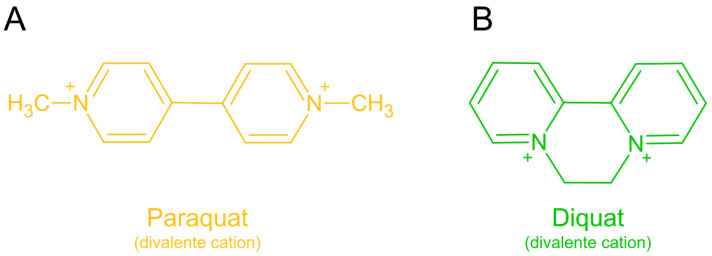
Chemical structures of the bipyridinium family of herbicides. (**A**) 1,1′-dimethyl-4,4′-bipyridilium (paraquat) divalent cation. (**B**) 1,1′-ethylene-2,2′-bipyridyldiylium (diquat) divalent cation. The illustration was prepared using Microsoft PowerPoint 2021, and the chemical structures were drawn using ChemSketch freeware 2023.2.4 [7].

**Figure 2 antioxidants-13-01354-f002:**
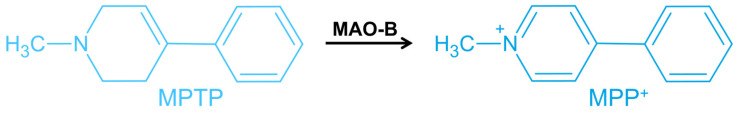
Chemical structures of MPTP and MPP^+^. The 1-methyl-4-phenyl-1,2,3,6-tetrahydropyridine (MPTP) is the precursor of the neurotoxin 1-methyl-4-phenylpyridinium (MPP^+^), which inhibits complex I of the mitochondrial electron transport chain. The metabolic conversion of MPTP to MPP^+^ is catalyzed by MAO-B present in astrocytes. MPP^+^ is translocated by the dopamine transporter, specifically causing degeneration of dopaminergic neurons in the *substantia nigra* and is utilized as a model of Parkinson’s disease. The illustration was prepared using Microsoft PowerPoint 2021, and the chemical structures were drawn using ChemSketch freeware 2023.2.4 [7].

**Figure 3 antioxidants-13-01354-f003:**
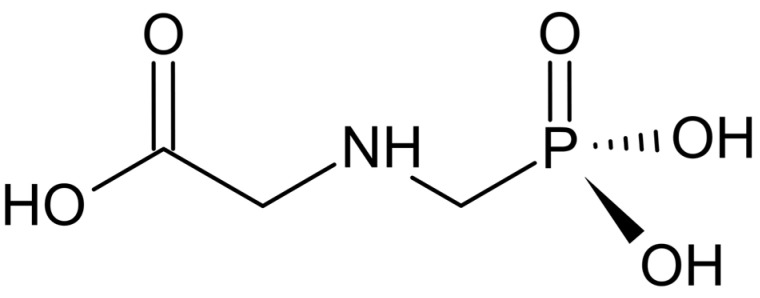
Chemical structure of glyphosate. Glyphosate inhibits the enzyme 5-enolpyruvylshikimate-3-phosphate synthase, which participates in the shikimate pathway responsible for the production of aromatic amino acids in plants. The chemical structure was drawn using ChemSketch freeware 2023.2.4 [7].

**Figure 4 antioxidants-13-01354-f004:**
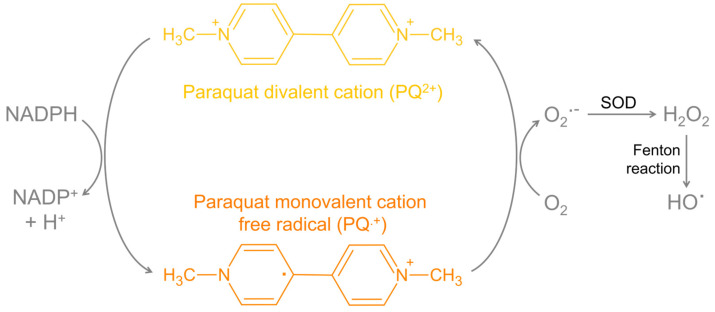
Redox cycle of paraquat. Paraquat divalent cation (PQ^2^⁺) can be reduced to paraquat monovalent cation free radical (PQ^•^⁺) by microglial nicotinamide adenine dinucleotide phosphate (NADPH) oxidase in the cytoplasm (in other organs, paraquat reduction is also catalyzed by NADPH-CYP450 reductase). This redox cycle consumes NADPH and generates superoxide radical (O_2_^•−^), which can be detoxified by superoxide dismutase (SOD) to hydrogen peroxide (H_2_O_2_). Via the Fenton reaction, H₂O₂ can be converted into the highly reactive hydroxyl radical (HO^•^). NADP, nicotinamide adenine dinucleotide phosphate oxidized form; O_2_, molecular oxygen. The illustration was prepared using Microsoft PowerPoint 2021, and the chemical structures were drawn using ChemSketch freeware 2023.2.4 [7].

**Figure 5 antioxidants-13-01354-f005:**
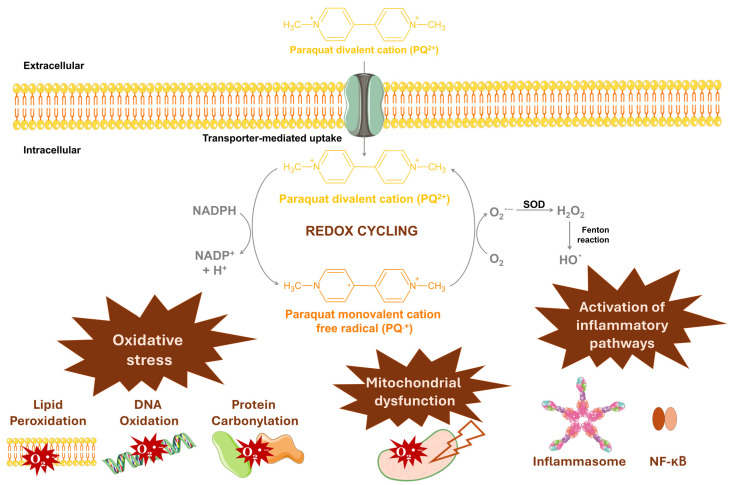
General mechanisms of paraquat toxicity: Paraquat divalent cation (PQ^2+^) can enter cells and undergo redox cycling, generating large amounts of reactive oxygen species, specifically superoxide radicals (O_2_^•−^), leading to oxidative stress. Additionally, paraquat can induce mitochondrial dysfunction by interfering with the electron transport chain and trigger systemic inflammatory responses. The illustration was prepared using Microsoft PowerPoint 2021 and images from Servier Medical Art. Chemical structures were drawn using ChemSketch freeware 2023.2.4 [7].

**Figure 6 antioxidants-13-01354-f006:**
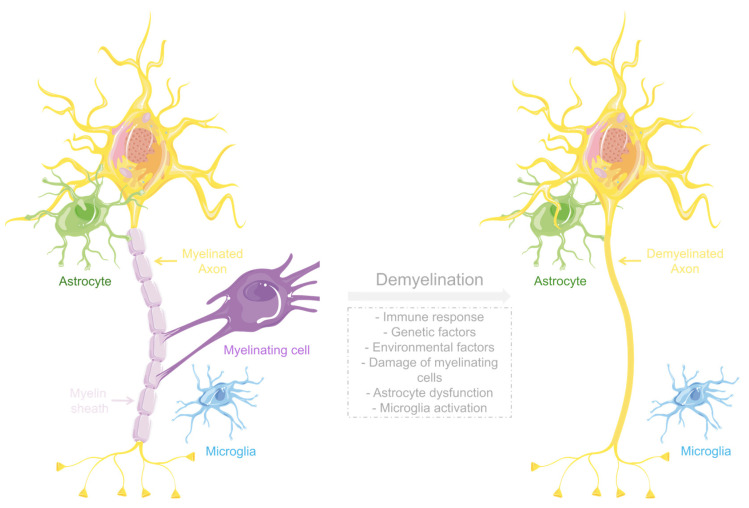
Pathophysiological mechanisms of demyelination. The illustration was prepared using Microsoft PowerPoint 2021 and images from Servier Medical Art.

**Figure 7 antioxidants-13-01354-f007:**
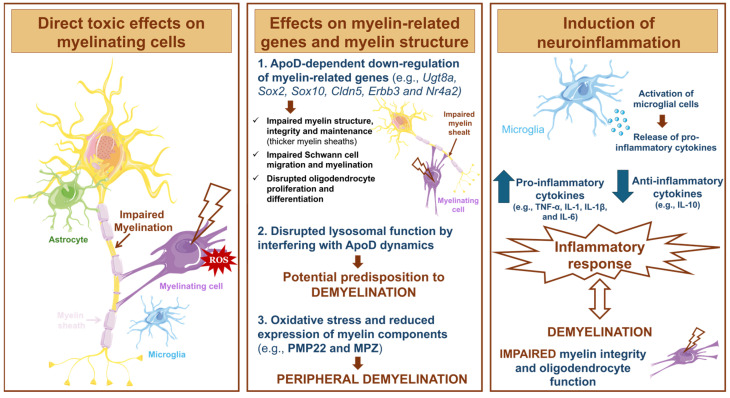
Effects of paraquat on myelination. Paraquat has been shown to induce direct toxic effects on myelinating cells, disrupt myelin structure and the expression of myelin-related genes, and cause neuroinflammation, contributing to demyelination. The illustration was prepared using Microsoft PowerPoint 2021 and images from Servier Medical Art.

## Data Availability

Not applicable.

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
