# Peer review of "The Link Between Paraquat and Demyelination: A Review of Current Evidence"

_antioxidants, 2024, doi:10.3390/antiox13111354_

Round 1

Reviewer 1 Report

I believe that a review should give a broad overview of the literature, and not limit itself to a few selected cases.

No comments

Author Response

REVIEWER 1

  1. Title: The title is ambiguous. Is it toxic or not?

Authors’ response: We greatly appreciate the Reviewer's comment. In line with this comment and a suggestion from the Editor, we have changed the title of the manuscript to “The link between paraquat and demyelination: A review of current evidence” We believe this title is more assertive and better reflects the content of the manuscript.

  1. Abstract/Introduction: As the title the abstract do not provide the necessary informations: In particular is non clear the point of view of the authors. In particular expressions as "Some evidences" should be avoided...

Authors’ response: We greatly appreciate the Reviewer's comments and suggestion. In line with these, we have revised the abstract to focus the manuscript on paraquat and demyelination by removing the general information on neurotoxicants (as it was also suggested by Reviewer 2). Moreover, we have improved our point of view about the goal of the manuscript and removed the expression “some evidence”.

The revised abstract reads as follows:

“Paraquat (1,1′-dimethyl-4,4′-bipyridilium dichloride), a widely used bipyridinium herbicide, is known for inducing oxidative stress, leading to extensive cellular toxicity, particularly in the lungs, liver, kidneys, and central nervous system (CNS), and is implicated in fatal poisonings. Due to its biochemical similarities with the neurotoxin 1-methyl-4-phenylpyridinium (MPP+), paraquat has been used as a Parkinson’s disease model, although its broader neurotoxic effects suggest the participation of multiple mechanisms. Demyelinating diseases are conditions characterized by damage to the myelin sheath of neurons. They affect the CNS and peripheral nervous system (PNS), resulting in diverse clinical manifestations. In recent years, growing concerns have emerged about the impact of chronic, low-level exposure to herbicides on human health, particularly due to agricultural runoff contaminating drinking water sources and their presence in food. Studies indicate that paraquat may significantly impact myelinating cells, myelin-related gene expression, myelin structure, and cause neuroinflammation, potentially contributing to demyelination. Therefore, demyelination may represent another mechanism of neurotoxicity associated with paraquat, which requires further investigation. This manuscript reviews the potential association between paraquat and demyelination. Understanding this link is crucial for enhancing strategies to minimize exposure and preserve public health”.

We hope the content of the revised abstract meets the Reviewers' expectations.

  1. References: I think that many references to scientific works are missing. Although I have no intention, for various reasons, of listing them.

Authors’ response: We greatly appreciate the Reviewer's comment. In line with this feedback and suggestions from other Reviewers, we have added references in some parts of the revised manuscript.

  1. Relevant contribution: In practice it reviews, without drawing relevant conclusions, a part of the literature.

Authors’ response: We greatly appreciate the Reviewer's comment. The goal of this manuscript is to collect information linking paraquat to demyelination. Only a few studies available in the literature report an association between paraquat and damage to myelinating cells, changes in myelin-related gene expression, alterations in myelin structure, and neuroinflammation. As such, due to the limited number of studies, no clear conclusions can be drawn. Therefore, we believe that this topic requires deeper investigation, as growing concerns are emerging about the impact of herbicides on human health, particularly due to agricultural runoff contaminating drinking water sources and their presence in food.

  1. I believe that a review should give a broad overview of the literature, and not limit itself to a few selected cases.

Authors’ response: We greatly acknowledge the Reviewer for the feedback on the manuscript. Our intention was to collect the available evidence in the literature linking paraquat to effects on myelin, myelin-related genes, or myelinating cells. Therefore, this is a very specific topic, which explains why only a few studies were selected. We hope the Reviewer understands our goal and perspective.

Reviewer 2 Report

Paraquat is a ever globally used herbicide, despite banned by many countries, some areas are still in use at present, due to its effectiveness. It well-known for its acute, high dosage toxicity, but its chronic, low level exposure is a more common case that potentially affect public health. In addition to lung and kidney toxicity, mounting evidences indicate that Paraquat can also be detrimental to oligodendrocyte and Schwann cells, consequently results in demylination. In-depth understanding how paraquat poison mylination cell has great implication for public health. This paper reviews recent research on paraquat induced mylination dysfunction at structural as well as molecular level, also indirect avenue such as glial cell mediated effects was included. Major issues were discussed and future perspectives were put forward. Overall this is comprehensive review on important topics, worth to read for student and researcher who are interested in this field. 

There are much contents irrelevant to the core theme, such as other herbicides and toxicants. I suggest curtail contents, focus on core topics. There are also much content that are common senses and can be seen in text books, I suggest cut off and increase leading edge materials. Since chronic, low level exposure is the major case that potentially affect public health, readers are likely interested in the epidemiology data concerning neuro-image, pathology as well as functional access in who exposed, or the animals around the areas paraquat in use, at least, this issue should be included in discussion or perspective.

Author Response

REVIEWER 2

Paraquat is a ever globally used herbicide, despite banned by many countries, some areas are still in use at present, due to its effectiveness. It well-known for its acute, high dosage toxicity, but its chronic, low level exposure is a more common case that potentially affect public health. In addition to lung and kidney toxicity, mounting evidence indicate that Paraquat can also be detrimental to oligodendrocyte and Schwann cells, consequently results in demyelination. In-depth understanding how paraquat poison myelination cell has great implication for public health. This paper reviews recent research on paraquat induced myelination dysfunction at structural as well as molecular level, also indirect avenue such as glial cell mediated effects was included. Major issues were discussed and future perspectives were put forward. Overall, this is comprehensive review on important topics, worth to read for student and researcher who are interested in this field. 

Authors’ response: We greatly acknowledge the Reviewer for the positive feedback on the manuscript.

  1. Title: There are much contents irrelevant to paraquat and demyelination, rendering this manuscript not sharply focus.

Authors’ response: We greatly appreciate the Reviewer's comment. In line with this comment and a suggestion from the Editor, we have changed the title of the manuscript to “The link between paraquat and demyelination: A review of current evidence”. We believe this title is more assertive and better reflects the content of the manuscript. Moreover, we have revised the abstract and introduction to focus the manuscript on paraquat and demyelination by removing the general information on neurotoxicants and other herbicides (Please see also the comment below).

  1. Abstract/Introduction: Curtail contents to focus on paraquat and demyelination, increase information concerning major issues.

Authors’ response: We greatly appreciate the Reviewer's suggestions. In line with these suggestions, we have revised the abstract and introduction to focus the manuscript on paraquat and demyelination by removing the general information on neurotoxicants. Moreover, we have improved the information regarding the major issues associated with herbicide use.

The revised abstract reads as follows:

“Paraquat (1,1′-dimethyl-4,4′-bipyridilium dichloride), a widely used bipyridinium herbicide, is known for inducing oxidative stress, leading to extensive cellular toxicity, particularly in the lungs, liver, kidneys, and central nervous system (CNS), and is implicated in fatal poisonings. Due to its biochemical similarities with the neurotoxin 1-methyl-4-phenylpyridinium (MPP+), paraquat has been used as a Parkinson’s disease model, although its broader neurotoxic effects suggest the participation of multiple mechanisms. Demyelinating diseases are conditions characterized by damage to the myelin sheath of neurons. They affect the CNS and peripheral nervous system (PNS), resulting in diverse clinical manifestations. In recent years, growing concerns have emerged about the impact of chronic, low-level exposure to herbicides on human health, particularly due to agricultural runoff contaminating drinking water sources and their presence in food. Studies indicate that paraquat may significantly impact myelinating cells, myelin-related gene expression, myelin structure, and cause neuroinflammation, potentially contributing to demyelination. Therefore, demyelination may represent another mechanism of neurotoxicity associated with paraquat, which requires further investigation. This manuscript reviews the potential association between paraquat and demyelination. Understanding this link is crucial for enhancing strategies to minimize exposure and preserve public health”.

The revised introduction reads as follows:

“Pesticides, particularly herbicides, are widely used chemicals commonly applied in agriculture to protect crops from pests. Herbicides can be grouped into distinct classes according to their mechanisms of action. Thus, depending on the class, herbicides may induce a specific set of adverse effects in humans, especially in cases of acute poisoning [1].

Paraquat (1,1′-dimethyl-4,4′-bipyridilium dichloride) is a non-selective contact herbicide of the bipyridinium chemical family (Figure 1A), classified as a photosystem I electron diverter and widely used worldwide. Paraquat's toxicity primarily results from its ability to induce oxidative stress. Upon entering biological systems, paraquat undergoes redox cycling, generating reactive oxygen species (ROS) such as superoxide radicals (O2•−). These ROS cause extensive cellular toxicity through lipid peroxidation, protein oxidation, and DNA damage [2-4]. The lungs are particularly susceptible to paraquat harmful effects due to its accumulation in alveolar cells, leading to pulmonary fibrosis and respiratory failure [5, 6]. Additionally, paraquat can affect other organs, including the liver, kidneys, and central nervous system (CNS), contributing to its overall systemic toxicity [6].

Due to the molecular and biochemical similarities of paraquat and the 1-methyl-4-phenylpyridinium (MPP+; also known as cyperquat), the active form of the neurotoxin 1-methyl-4-phenyl-1,2,3,6-tetrahydropyridine (MPTP; Figure 2), which causes Parkinson-like symptoms in both animals and humans, paraquat has been increasingly investigated as a model of Parkinson’s disease [8, 9]. However, as seen for several neurotoxicants, the effects of paraquat on the nervous system may result from a multiplicity of targets and mechanisms [2, 4, 8, 9]. Thus, the impact of paraquat on the nervous system is expected to be profound.

Demyelinating diseases represent a large group of pathologies characterized by defects in myelin sheaths that surround neuronal axons. Among this group of diseases are multiple sclerosis (MS), the most prevalent condition, neuromyelitis optica, acute disseminated encephalomyelitis, transverse myelitis, Guillain-Barré syndrome, chronic inflammatory demyelinating polyneuropathy, and central pontine myelinolysis (CPM), among others. These diseases can affect neurons in the CNS and peripheral nervous system (PNS), resulting in a wide spectrum of clinical manifestations with variable ages of onset, progression, and recovery [10-14]. The exact etiology of these pathologies remains elusive, although genetic predisposition, autoimmune mechanisms, and environmental factors are believed to play pivotal roles.

In recent years, growing concerns have emerged regarding the impact of chronic, low-level exposure to herbicides on human health. This issue is particularly serious due to agricultural runoff, which contaminates drinking water sources. Moreover, the presence of these chemicals in food supplies may pose significant effects to human health. While the connection between demyelination and the development of demyelinating diseases has been recognized for some herbicides, including glyphosate (5-enolpyruvylshikimate-3-phosphate synthase inhibitor) [15] and diquat (Figure 1B; belongs to the same chemical family as paraquat) [16, 17], it remains less evident for paraquat, despite its other well-known effects on the nervous system.

This manuscript reviews the available information linking paraquat and demyelination. Understanding this putative link could lead to more effective public health policies and regulations, as well as help develop targeted strategies to minimize exposure and associated risks. This knowledge could have broader implications for understanding the neurobiology of demyelinating diseases and may also potentially inform research on other neurological diseases and conditions”.

We hope the content of the revised abstract and introduction meets the Reviewers' expectations.

  1. There are much contents irrelevant to the core theme, such as other herbicides and toxicants. I suggest curtail contents, focus on core topics. There are also much content that are common senses and can be seen in text books, I suggest cut off and increase leading edge materials. Since chronic, low level exposure is the major case that potentially affect public health, readers are likely interested in the epidemiology data concerning neuro-image, pathology as well as functional access in who exposed, or the animals around the areas paraquat in use, at least, this issue should be included in discussion or perspective.

Authors’ response: We greatly appreciate the Reviewer's suggestions. In line with this and the previous comment, we have revised the introduction to focus the manuscript on paraquat and demyelination by removing the general information on neurotoxicants. We also removed the information related to the other herbicides on section 6 to make the manuscript more focused on paraquat.

We also improved the information regarding the major issues associated with herbicide use. In the revised manuscript, it reads as follows:

“In recent years, growing concerns have emerged regarding the impact of chronic, low-level exposure to herbicides on human health. This issue is particularly serious due to agricultural runoff, which contaminates drinking water sources. Moreover, the presence of these chemicals in food supplies may pose significant effects to human health”.

Regarding the epidemiological data on neuroimaging, pathology, and functional assessments in individuals or animals exposed to paraquat, no current data associates paraquat exposure with demyelination. This lack of information may stem from either the absence of studies on this subject or from the failure to detect any association. However, we believe that the absence of investigation is more likely due to the limited information available on this potential link. Therefore, based on the Reviewer’s suggestion, we have expanded the future perspectives section to emphasize that such studies could be valuable in elucidating the consequences of chronic, low-level paraquat exposure, particularly its role in inducing demyelination. In the revised manuscript, it reads as follows:

“On the other hand, the available in vivo studies primarily focus on acute exposures to relatively high concentrations of paraquat. This experimental design may enhance the effects of paraquat, allowing a more precise elucidation of the underlying mechanistic pathways. However, it may not accurately represent scenarios of prolonged exposure to low levels of the herbicide needed to evaluate its cumulative effects. Thus, longitudinal epidemiological studies are needed to analyze populations chronically exposed to low levels of paraquat for extended periods, focusing on cognitive and motor function assessments. Neurological examinations to identify signs of demyelination or other neurodegenerative conditions, particularly in agricultural communities where paraquat use is prevalent, would also be beneficial. Moreover, advancements in pathology, and functional assessments will be also key in understanding the long-term impact of paraquat exposure on humans and animals in affected areas. Such knowledge is important for establishing any causal relationship between paraquat exposure and long-term effects on the CNS and PNS. Therefore, more studies are needed to elucidate the potential impact of paraquat on myelination and neuroinflammation over time”.

We hope these alterations meet the Reviewer’s expectations.

Reviewer 3 Report

the manuscript is pertinent to the journal and overall is written clearly

The topic is of interest to a broad audience given the widespread environmental exposures.

The basic discussion on oligodendrocytes [section 3] is too long and a bit off-topic. It should be condensed and shortened as none of the concepts are needed to understand the key points in the manuscript.

The sentences from Line 317 require references.

It would help if the writers could suggest potential means of detoxifying paraquat and other herbicidal contaminants on foods. If it's not possible--then stating that would also be useful. Please also clarify if organic-designation means no paraquat exposure/contamination.

the manuscript is pertinent to the journal and overall is written clearly

The topic is of interest to a broad audience given the widespread environmental exposures.

The basic discussion on oligodendrocytes [section 3] is too long and a bit off-topic. It should be condensed and shortened as none of the concepts are needed to understand the key points in the manuscript.

The sentences from Line 317 require references.

It would help if the writers could suggest potential means of detoxifying paraquat and other herbicidal contaminants on foods. If it's not possible--then stating that would also be useful. Please also clarify if organic-designation means no paraquat exposure/contamination.

Author Response

REVIEWER 3

The manuscript is pertinent to the journal and overall is written clearly. The topic is of interest to a broad audience given the widespread environmental exposures

Authors’ response: We greatly acknowledge the Reviewer for the positive feedback on the manuscript.

  1. The basic discussion on oligodendrocytes [section 3] is too long and a bit off-topic. It should be condensed and shortened as none of the concepts are needed to understand the key points in the manuscript.

Authors’ response: We greatly appreciate the Reviewer's suggestion. In line with this, we have condensed and shortened the information in Section 3, keeping only the essential details related to myelin and its formation so that readers can understand its importance in neuronal function. We hope these alterations meet the Reviewer’s expectations.

  1. The sentences from Line 317 require references.

Authors’ response: We greatly appreciate the Reviewer's comment. After restructuring and condensing the information in Section 3 (as per the previous comment), the sentences from line 317 in the original manuscript were removed in the revised version. Therefore, this comment has been inherently addressed.

  1. It would help if the writers could suggest potential means of detoxifying paraquat and other herbicidal contaminants on foods. If it's not possible--then stating that would also be useful. Please also clarify if organic-designation means no paraquat exposure/contamination.

Authors’ response: We greatly appreciate the Reviewer's suggestion and comment. In line with the Reviewer’s suggestion, we provide in the future perspective section some potential alternatives to detoxify paraquat and other herbicidal contaminants on foods. In the revised manuscript, it reads as follows:

“Since food contamination with herbicides may contribute to chronic low-level exposure to these chemicals, improving strategies for detoxifying paraquat and other herbicidal contaminants in food is an important area of research. Several potential methods for reducing or eliminating herbicide residues in food have been explored. These strategies include: (1) washing fruits and vegetables under running water to remove some surface residues of herbicides like paraquat [148]; (2) peeling fruits or vegetables with thicker skins, such as apples, potatoes, or cucumbers, to significantly reduce herbicide residue, and removing outer leaves of leafy vegetables where residues may accumulate [149]; (3) soaking in salt or vinegar solutions [148]; and (4) using of cooking methods, such as boiling, blanching, and stir-frying, can break down certain herbicide residues [150]. These methods can help limit exposure to herbicides from contaminated food”.

In Section 2, when we state that “…and the use of organic compounds”, we are referring to materials such as ashes or animal manure. In this context, it does not imply paraquat exposure or contamination, as no chemicals had been used at that time.

We hope these alterations and clarification meet the Reviewer’s expectations.

Round 2

Reviewer 2 Report

The manuscript is much better by addition of more information about paraquat on mylination, which is the core point of your topic.Despite it is overall good , some minior issues may need to add or revised. 

I would like you consider to add some more materials make this review more insightful.

1. As you mentioned, paraquat prefer to accumulate in lung, as well as demylination maybe an indirect effect. Emerging evidence indicated that brain function and pathology has relation with lung. I suggest you add some research findings, or at least talk it in future perspectives section.

2. Since present findings suggest ApoD is a key mediator for demylination, it would be better make readers clear which cell type express ApoD and which cell type can be affect by ApoD;

3. Single cell is an emerging powerful tool for dissecting pathogenesis at single cell (cell type) level, including not only mRNA sequence. I would like you add some references and findings on neural diseases utilizing single cell sequence to exemplify(line 677).

4. As high-dosage acute paraquat intake can lead to PD like symptom, most demylination research in paraquat induce animal model lack attention on mid-brain substantia nigra where is PD core pathological zone. This concern should be mention at least in future perspectives section.

Author Response

REVIEWER 2

The manuscript is much better by addition of more information about paraquat on myelination, which is the core point of your topic. Despite it is overall good , some minor issues may need to add or revised. 

Authors’ response: We greatly acknowledge the Reviewer for the positive feedback on the revised manuscript.

  1. As you mentioned, paraquat prefer to accumulate in lung, as well as demyelination maybe an indirect effect. Emerging evidence indicated that brain function and pathology has relation with lung. I suggest you add some research findings, or at least talk it in future perspectives section.

Authors’ response: We greatly appreciate the Reviewer's suggestion. In line with this suggestion, in the revised manuscript, we have explored the relationship between lung health and brain (dys)function in the future perspectives' section as an hypothesis regarding paraquat’s effects on the CNS, including on myelin and myelinating cells.

In the revised manuscript, it reads as follows:

“Studies have suggested a significant relationship between lung health and brain function and pathology [152, 153]. Lung function can influence brain processes, potentially due to the brain dependency on efficient blood oxygenation [154]. Thus, poor lung function, often associated with low oxygen levels and inflammation, is linked to cognitive decline and accelerates the progression to dementia [153, 155]. Particularly, chronic respiratory diseases, such as chronic obstructive pulmonary disease and asthma, have been correlated with increased risk of neuroinflammation, oxidative stress, and even Alzheimer's disease [156-158]. Considering that lungs are especially vulnerable to the harmful effects of paraquat, which accumulates in alveolar cells, causing pulmonary fibrosis and eventually respiratory failure [5, 6], it can be hypothesized that such lung damage might contribute to paraquat’s effects on the CNS, including on myelin and myelinating cells. However, this relationship remains unclear, and future research should aim to uncover the pathways involved in paraquat-induced toxicity in myelinating cells, including this putative relationship”.

We hope these alterations meet the Reviewer’s expectations.

  1. Since present findings suggest ApoD is a key mediator for demyelination, it would be better make readers clear which cell type express ApoD and which cell type can be affect by ApoD;

Authors’ response: We greatly appreciate the Reviewer's suggestions. In line with these suggestions, in the revised manuscript, we have clarified which nervous system cell types express ApoD and which cell populations may be affected by altered ApoD expression and/or function.

In the revised manuscript, it reads as follows:

“Apolipoprotein D (ApoD) is a lipocalin lipid-binding protein expressed in oligodendrocytes [100], Schwann cells [101], pericytes [102], and astrocytes [103], and possesses antioxidant properties. Although mature neurons do not express ApoD, they are capable of internalizing this protein from the surrounding extracellular environment, especially under conditions of oxidative stress [104]. Thus, altered ApoD expression and/or function are likely to impact both the producing and the target cells, including neurons.

Studies in cultured cells and animal models have shown that ApoD is essential for nervous system homeostasis and normal function, as well as for the development and preservation of crucial neural structures [105]. Specially, ApoD is involved in various myelin-related functions, including myelin maintenance, myelin clearance following axonal injury, and the negative regulation of macrophage recruitment to damaged axonal areas [106]. Moreover, in the absence of ApoD, axon regeneration and remyelination are delayed, and reduced myelin sheath thickness is detected even in uninjured nerves [106]”.

We hope these alterations meet the Reviewer’s expectations.

  1. Single cell is an emerging powerful tool for dissecting pathogenesis at single cell (cell type) level, including not only mRNA sequence. I would like you add some references and findings on neural diseases utilizing single cell sequence to exemplify (line 677).

Authors’ response: We greatly appreciate the Reviewer's suggestion. In line with this suggestion, we have expanded the text to include a variety of single-cell approaches, such as single-cell and single-nucleus RNA sequencing, ATAC sequencing, single-cell epigenomics, single-cell proteomics, and multiomics at the single-cell level. These tools may be particularly useful in studying alterations in specific cell populations associated with paraquat exposure. We have also added references to these advanced techniques, which now better illustrate their relevance in neurodegenerative conditions.

In the revised manuscript, it reads as follows:

“At this level, single-cell and single-nucleus RNA sequencing [159], single-cell and single-nucleus assay for transposase-accessible chromatin (ATAC) sequencing [160], single-cell epigenomics [161], single-cell proteomics [162], or multiomics at the single‐cell level [163] may provide valuable insights into changes within specific cell populations associated with paraquat exposure. These advanced techniques have been successfully employed to examine cell-specific phenotypes and disrupted cellular processes, as myelination, in neurodegenerative and neuroinflammatory disease models, including Alzheimer’s disease, Parkinson’s disease, and MS, holding significant promise for applications in neurotoxicology studies [159]”.

We hope these alterations meet the Reviewer’s expectations.

  1. As high-dosage acute paraquat intake can lead to PD-like symptom, most demyelination research in paraquat induce animal model lack attention on mid-brain substantia nigra where is PD core pathological zone. This concern should be mention at least in future perspectives section.

Authors’ response: We greatly appreciate the Reviewer's suggestion. In line with this suggestion, we have improved the future perspectives section to highlight the need for research focused on the substantia nigra in paraquat-exposed animals, as this is the key pathological region affected in Parkinson’s disease. Thus, since paraquat is known to induce Parkinson-like symptoms, and is commonly used as a model for this disease, examining the potential impact of paraquat on myelin and myelinating cells in this brain region is particularly relevant.

In the revised manuscript, it reads as follows:

“On the other hand, the available in vivo studies primarily focus on acute exposures to relatively high concentrations of paraquat [97, 115, 164]. This experimental approach may amplify the effects of paraquat, enabling a more precise elucidation of the underlying mechanistic pathways behind its toxic effects. Notably, exposure to a high dose of paraquat can induce Parkinson’s disease-like symptoms [4]. Thus, future research examining the substantia nigra (the key pathological brain region affected in Parkinson’s disease [165]) in paraquat-exposed animals could provide valuable insights into paraquat's effects on myelin and myelinating cells”.

We hope these alterations meet the Reviewer’s expectations.
